# Hidden in Order: A Theoretical and Empirical Dissection of Positional Bias in LLMs

## Abstract

Large language models (LLMs) demonstrate strong capabilities but remain limited by positional bias defined as the tendency to treat content differently based on where it appears in a sequence. This bias creates subtle but consequential distortions in tasks that demand equal treatment of all inputs, such as multi-document reasoning. Although positional effects have been observed, no unified approach has existed to measure them systematically. We present Xayna, a framework of 11 complementary methods spanning information-theoretic, geometric, and probabilistic perspectives. Xayna evaluates internal representations, attention flows, and output behavior across models including GPT-4o, Gemini 2.5, Llama 3, Claude 3.7, Mixtral 8×22B, and DeepSeek R1. Results show consistent patterns: (i) identical content yields divergent representations with near-zero cosine similarity; (ii) attention flows disproportionately from later to earlier tokens; and (iii) primacy dominates behavior, with position-1 preferences reaching 0.892 in Llama 3. Xayna provides both a toolkit for quantifying positional bias and a foundation for developing position-aware evaluation and mitigation strategies, advancing the pursuit of more reliable and equitable language technologies.

## 1 Introduction

Large language models (LLMs) have achieved remarkable success in natural language understanding and generation (Bommasani et al., 2021). However, their behavior often diverges from rational information processing. Positional bias, while recognized in prior studies, has largely been examined in isolated contexts rather than through a systematic framework. It refers to the tendency to assign different importance to information based solely on where it appears in a sequence (Zhao et al., 2021; Liu et al., 2023b).

This bias is especially problematic in tasks where order should not matter, such as multi-document QA, summarization, or comparative analysis Zheng et al. (2024); Yu et al. (2025). In these settings, relevance ought to be determined by content rather than placement, yet positional bias undermines fairness, reliability, and robustness (Bender et al., 2021). For instance, in legal contexts, evidence should not carry different weight depending on when it appears, and in a literature review, synthesis should not be distorted by whether papers are listed chronologically or alphabetically.

Unlike visible social biases, positional effects often arise subtly from model architecture and training dynamics, making them harder to detect. Previous work has identified specific symptoms recency effects in masked LMs (Zhao et al., 2021), order effects in a few shot prompting (Liu et al., 2021), but no comprehensive framework has systematically captured their scope.

Beyond fairness, positional bias also impacts efficiency and scalability. Agentic systems, where LLMs perform multi-step reasoning via APIs, tools, or chained sub-queries, are already resource-intensive. Commercial APIs incur high costs, while open-source deployments demand large-scale inference. When outputs shift with superficial reordering, developers must issue redundant queries, permutations, or self-consistency runs to stabilize results. This compounds overhead and undermines reproducibility, making positional bias both a scientific and practical bottleneck.

These challenges underscore the need for principled methods to rigorously characterize positional bias. To address this, we introduce Xayna as the first systematic and extensible framework that detects, quantifies, and analyzes positional bias in LLMs. By treating positional bias as a set of

interrelated effects across computational stages, Xayna integrates eleven complementary methods that together provide representational, mechanistic, and behavioral perspectives (see Section 3)

Applying Xayna to contemporary LLMs with semantically equivalent but reordered document pairs, we find that positional biases are pervasive, statistically significant, and manifest at all levels of operation. Key results show that (i) identical content yields divergent internal representations, (ii) attention mechanisms consistently privilege earlier content, (iii) Bayesian analyses reveal strong primacy effects, and (iv) reordering documents frequently changes outputs.

Together, these findings establish positional bias as a systemic property of transformer architectures. By providing rigorous tools for quantification and comparison, Xayna lays a foundation for deeper theoretical understanding and for principled mitigation strategies, advancing the development of language technologies that are more robust, equitable, and position-invariant.

## 2 RELATED WORK

Transformers lack recurrence or convolution and therefore rely on explicit positional encodings (PEs) to model order (Vaswani et al., 2017). Sinusoidal and learned absolute encodings (Devlin et al., 2019) were later extended to relative forms (Shaw et al., 2018; Dai et al., 2019; Raffel et al., 2020) and rotational schemes such as RoPE (Wang et al., 2024). While these approaches improved scaling and context length, they also introduced structural limitations: poor extrapolation beyond training lengths (Press et al., 2022; Kazemnejad et al., 2024) and systematic primacy/recency preferences. Prior surveys catalog these weaknesses (Kazemnejad et al., 2024), but they remain treated as isolated quirks of encoding design. Our work takes these architectural roots as motivation for a broader claim: positional bias is not a corner case of encoding, but a fundamental property of how transformers process order information.

Recent studies have documented position-driven effects across diverse applications. In long-context modeling, LLMs fail when relevant content appears mid-sequence ("lost in the middle" (Liu et al., 2023a)). In evaluation, outputs flip with candidate order (Wang et al., 2023), and multiple-choice QA shows sharp sensitivity to option placement (Pezeshkpour & Hruschka, 2023; Zheng et al., 2024). Cognitive-bias benchmarks capture similar primacy and recency effects (Ye et al., 2024), and multi-agent judging setups amplify them further (Ma et al., 2025). These studies demonstrate that positional bias pervades reasoning, retrieval, and evaluation, however each focuses on a narrow symptom. In contrast, we aim to propose a framework (Xayna) which consolidates these disparate observations under a single framework, allowing systematic comparison across tasks and levels of analysis.

Several works target specific manifestations of positional bias. Notably mitigation strategies include PriDe which removes token-ID bias in MCQs (Zheng et al., 2024), PINE that re-engineers causal attention and RoPE to eliminate evaluation order effects (Wang et al., 2025), and hidden-state scaling methods reduce long-context failures (Yu et al., 2025). While effective, these are point solutions that address local failures without providing a general diagnostic view. Our aim is to offer the first comprehensive framework that spans representational geometry, attention dynamics, and probabilistic outputs. By unifying eleven complementary methods, it reveals positional bias as a systemic architectural property and provides actionable metrics that can guide principled mitigation, whether through data augmentation, regularization losses, or architectural design. Unlike these mitigation-focused approaches, Xayna does not directly intervene but instead diagnoses where and how positional bias manifests, thereby informing which mitigation strategies are most appropriate to apply.

## 3 THE XAYNA FRAMEWORK: METHODOLOGIES FOR POSITIONAL BIAS DISSECTION

The Xayna framework (shown in Fig. 1) is built on the notion that positional bias is not a single effect but a collection of related phenomena. No single metric can capture its full range. Xayna therefore integrates 11 complementary methodologies, each probing a different facet of how positional information shapes LLM behavior, including representational geometry, attention dynamics, output

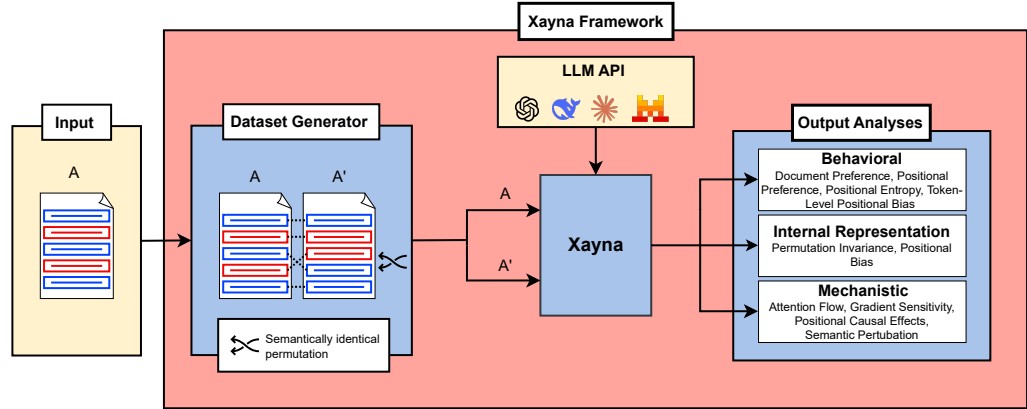

Figure 1: Overview of the Xayna framework. The input is a document $A$ and its semantically equivalent variant $A'$, generated with controlled lexical and syntactic alterations. Both are passed through LLMs and analyzed within Xayna, which outputs behavioral, representational, and mechanistic measures that together characterize positional bias.

probabilities, and final predictions. These methods are grounded in established principles from information theory, geometric data analysis, causal inference, and Bayesian statistics.

To ensure comprehensive coverage, the methods are grouped into three categories: behavioral and output-level analyses, internal representation analyses, and mechanistic & causal analyses. Their complementarity is empirically validated by the low correlations between metrics (Fig. 4b), which show that each captures unique information and rules out redundancy. This breadth makes clear that positional bias cannot be understood from a single perspective, making a holistic approach essential. Providing such an approach is a central contribution of our work.

To establish a clear and consistent notation, we define an input sequence as $S = (D_1, D_2, \ldots, D_N)$, representing an ordered concatenation of $N$ documents. In our primary experimental setup, we use pairs of semantically equivalent documents, $A$ and its variant $A'$, making a typical sequence $S_1 = (A, A')$ or its permutation $S_2 = (A', A)$. We introduce a precise notation for the hidden state representation of a specific document: $h(D, p, S)$ refers to the matrix of hidden state vectors for all tokens belonging to document $D$ when it is at position $p$ in sequence $S$. For methods requiring a single vector, we typically use the mean-pooled representation of these token vectors. We analyze how the model's processing $f(\cdot)$ and output $Y$ vary under different permutations of $\pi$ denote a permutation of $(1, \ldots, N)$, and $S_\pi = (D_{\pi(1)}, D_{\pi(2)}, \ldots, D_{\pi(N)})$ represent a specific ordering presented to the LLM. $S$, where $Y_\pi = f(S_\pi)$ may refer to probabilities, representations, or final predictions depending on the method. Pseudo-code for each method is provided in Appendix A.3, and detailed summaries are in Appendix A.1, Table 3.

### 3.1 BEHAVIORAL & OUTPUT-LEVEL ANALYSIS

**Document Order Preference Analysis (DOPA)**: DOPA directly measures behavioral bias by forcing the model to choose between two semantically equivalent documents presented in different orders. For the sequences $S_1 = (A, A')$ and $S_2 = (A', A)$, we prompt the model with a task like "which document is more informative?". Let $C(S)$ be the model's choice (position 1 or 2). The Positional Preference Score (PPS) for a position $i$ is the fraction of times it is chosen: $\text{PPS}(i) = \frac{\sum_k \mathbb{I}(C(S_k)=i)}{\text{Total trials}}$. Deviations from $0.5$ (for binary choice) indicate bias. Figure 2 illustrates DOPA preferences.

**Stochastic Prompt Shuffling & Bayesian Positional Bias Estimation (SPS-BPBE)**: This method extends DOPA by modeling the preference for a position as a probabilistic parameter. Over many trials involving different document pairs and random permutations, we record whether the document in position 1 is chosen. This series of choices is modeled as a Bernoulli process, $Y_m \sim \text{Bernoulli}(\theta_{\text{pos1}})$, where $\theta_{\text{pos1}}$ is the underlying preference for the first position. Using a Beta prior, $\theta_{\text{pos1}} \sim \text{Beta}(\alpha_0, \beta_0)$, we update our belief given the data. If position 1 is chosen $k$ times in $M$ trials, the posterior is

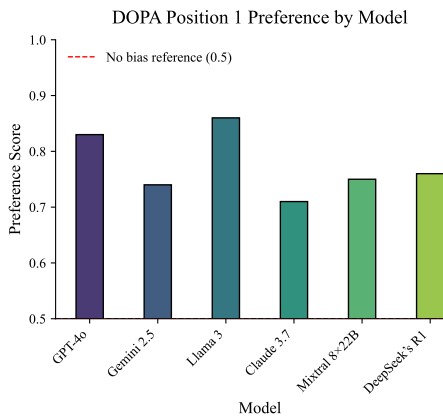 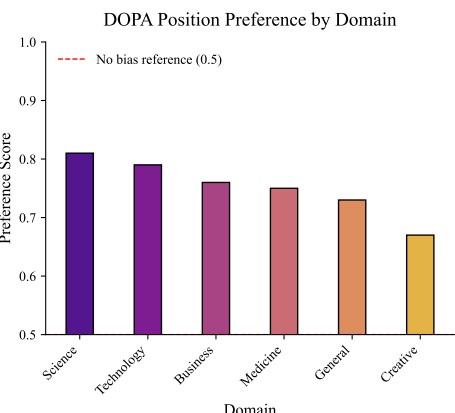

Figure 2: (a) Illustrative DOPA preference for content based on position across different LLMs. Higher bars indicate stronger preference for that position. (b) Illustrative breakdown of DOPA positional preference by document domain, showing domain-specific variations in bias.

$\theta_{\text{pos1}}|\text{data} \sim \text{Beta}(\alpha_0 + k, \beta_0 + M - k)$. The expectation $\mathbb{E}[\theta_{\text{pos1}}|\text{data}]$ provides a robust estimate of the model's primacy bias.

**Positional Entropy Analysis (PEA)**: PEA measures how reordering documents affects output uncertainty. It compares the entropy, denoted by $\mathcal{E}$ to avoid confusion with hidden states, of the output probability distributions from sequences $S_1 = (A, A')$ and $S_2 = (A', A)$. The entropy for a sequence $S$ is $\mathcal{E}(Y|S) = -\sum_y P(y|S) \log P(y|S)$. The bias metric is the absolute difference in entropy: $\Delta\mathcal{E} = |\mathcal{E}(Y|S_1) - \mathcal{E}(Y|S_2)|$. A large $\Delta\mathcal{E}$ indicates that position significantly alters the model's confidence or uncertainty in its predictions.

Table 1: DPAF: Mean inter-document attention flows ($F_{P1 \rightarrow P2}$ vs. $F_{P2 \rightarrow P1}$). P1 is the first document, P2 is the second. Higher $F_{P2 \rightarrow P1}$ indicates later content attends more to earlier content.

| Flow Direction | GPT-4o | Gemini 2.5 | Llama 3 | Claude 3.7 | Mixtral 8×22B | DeepSeek's R1 |
|---|---|---|---|---|---|---|
| $F_{P1 \rightarrow P2}$ | 0.254 | 0.251 | 0.252 | 0.247 | 0.254 | 0.252 |
| $F_{P2 \rightarrow P1}$ | 0.274 | 0.264 | 0.276 | 0.264 | 0.276 | 0.273 |

**Token-level Positional Bias Detection (TPBD)**: TPBD examines how document order affects the model's generated text at the token level. Given a query $Q$ and two sequences, $S_1 = (A, A')$ and $S_2 = (A', A)$, we compare the conditional probability distributions for generating a specific answer. The bias is measured by the Kullback-Leibler (KL) divergence between the output distributions under the two orderings: $\text{KL}(P(\text{Answer}|S_1, Q)||P(\text{Answer}|S_2, Q))$. A high KL divergence indicates that the position of the documents significantly skews the likelihood of generating certain tokens.

## 3.2 Internal Representation Analysis

**Permutation-Invariant Representation Analysis (PIRA)**: PIRA probes if the internal representation of identical content changes based on its position. For sequences $S_1 = (A, A')$ and $S_2 = (A', A)$, PIRA measures the dissimilarity between the representation of document $A$ when it appears in the first position (in $S_1$) versus the second position (in $S_2$). This is done by comparing $h(A, 1, S_1)$ with $h(A, 2, S_2)$. Centered Kernel Alignment (CKA) (Kornblith et al., 2019) is applied between the corresponding matrices of token representations, which we denote $H_A^{(1)} = h(A, 1, S_1)$ and $H_A^{(2)} = h(A, 2, S_2)$: $\text{CKA}(H_A^{(1)}, H_A^{(2)}) = \frac{\text{HSIC}(K_1, K_2)}{\sqrt{\text{HSIC}(K_1, K_1)\text{HSIC}(K_2, K_2)}}$. Low CKA (Figure 3a) indicates strong positional influence, suggesting that the model's internal "understanding" of the same information changes with order.

**Contrastive Representation Learning for Bias Detection (CRLBD)**: CRLBD trains a simple probe classifier, $g : \mathcal{R} \rightarrow \{p_1, \ldots, p_N\}$, to predict a document's original position $p$ from its hidden state

representation $h(D, p, S)$. The input space $\mathcal{R}$ is the space of the model's representations. High accuracy (Figure 3b) implies strong, separable positional encoding within the representations. This is often framed using an InfoNCE-style loss, pushing representations of the same content from different positions apart if the model encodes position, or together if a contrastive task aims to **remove** such bias.

**Frequency-Domain Positional Bias Analysis (FPBA)**: FPBA analyzes DFT$(H)[k]$ of token representations $H$, where $H$ corresponds to the matrix of token representations for a document (i.e., $h(D, p, S)$), to see if specific spectral components of representations change with document position. This can uncover biases in how periodic or structural positional information is encoded or utilized.

### 3.3 MECHANISTIC & CAUSAL ANALYSIS

**Dynamic Positional Attention Flow (DPAF)**: DPAF examines attention weights, which we denote as $\alpha_{jk}^{(l,h)}$ to avoid confusion with document A, between tokens to map information routing. For a sequence $S = (A, A')$, with corresponding token sets $T_A$ and $T_{A'}$, the flow from $A$ to $A'$ is calculated by aggregating attention across all layers and heads: $\text{Flow}(A \to A') = \sum_{l,h} w_{l,h} \left( \frac{1}{|T_A|} \sum_{j \in T_A} \sum_{k \in T_{A'}} \alpha_{jk}^{(l,h)} \right)$. We then compare the bidirectional flows, $\text{Flow}(A \to A')$ versus $\text{Flow}(A' \to A)$ (Table 1), where an asymmetry reveals preferential information integration.

**Positional Gradient Sensitivity (PGS)**: PGS computes $||\nabla_{e_p(i)} L||$ for loss $L$ and positional embedding $e_p(i)$. Comparing norms across positions (e.g., for identical content placed early vs. late) indicates whether the model's learning objective is disproportionately sensitive to positional signals from certain input segments, revealing biases in how optimization treats different positions.

**Causal Positional Intervention Analysis (CPIA)**: CPIA estimates the direct causal effect of a document's position on a specific outcome by performing counterfactual interventions. It calculates the Average Treatment Effect (ATE) of placing content $C$ in position $p_1$ versus $p_2$ on an outcome $Y$: $\text{ATE} = \mathbb{E}[Y|\text{do}(\text{pos}(C) = p_1)] - \mathbb{E}[Y|\text{do}(\text{pos}(C) = p_2)]$. This is achieved by directly modifying the model's internal mechanisms, such as swapping positional embeddings or altering attention patterns, to simulate these counterfactual scenarios and measure the resulting change in the outcome.

**Localized Semantic Perturbation (LSP)**: LSP quantifies how the position of a neighboring document influences the processing of a target document. In this setup, we have a target document $D_T$ containing the answer to a query $Q$, and a perturbing document $D_P$. We compare the model's performance on answering $Q$ across two sequences: $S_1 = (D_T, D_P)$ and $S_2 = (D_P, D_T)$. The bias is measured as the absolute difference of F1 score in the performance metric $\mathcal{M}$: $|\mathcal{M}(f(S_1, Q)) - \mathcal{M}(f(S_2, Q))|$. This score captures the distracting or context-altering effect caused by the neighbor's position (conceptualized in Figure 4a).

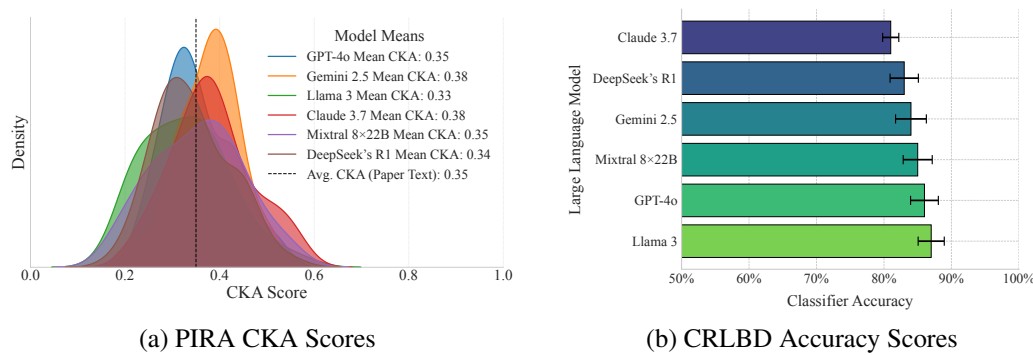

(a) PIRA CKA Scores          (b) CRLBD Accuracy Scores

Figure 3: (a) Illustrative distribution of PIRA CKA Scores, comparing representations of identical content at different positions. Lower CKA scores indicate greater dissimilarity due to position. (b) Illustrative CRLBD classification accuracy scores for predicting content position from its representation. Higher accuracy indicates stronger encoding of position.

## 4 DATASET AND EXPERIMENTAL SETUP

### 4.1 DATASET GENERATION

To rigorously investigate positional bias while controlling for content variability, we constructed a specialized dataset. This dataset comprises 1000 unique document pairs, generated programmatically using GPT-4o-mini across diverse domains: science (20%), medicine (20%), technology (15%), business (15%), general (15%), and creative (15%). Each "base" document was factual, coherent, and information rich, with a target length of 5000-7000 words, avoiding complex formatting or section headers to ensure structural uniformity. For each base document $A$, a corresponding variant $A'$ was created to be semantically almost identical to $A$ but with approximately 10-15% lexical and minor syntactic alterations (synonym substitutions, paraphrasing, reordering of clauses), preserving the core meaning and argumentative structure. The generation pipeline ensured that variations were distributed throughout the text and did not introduce new substantive information. The average semantic similarity between $A$ and $A'$, measured by the cosine similarity of their sentence-BERT embeddings, was maintained at $0.92 \pm 0.03(\sigma)$. The final dataset therefore contains 2000 documents (1000 pairs), with an average length of 5325 words ($\sigma = 87.3$).

Constructing the dataset synthetically ensured that content was carefully controlled, allowing position to serve as the only meaningful variable under manipulation and enabling a clean, rigorous analysis. Unlike simply permuting naturally generated text, which produces unnatural artifacts and distorts linguistic flow, our paired generation preserves semantic fidelity while maintaining naturalness, ensuring that any observed effects stem from position rather than text quality. Further details on the generation prompts and alteration methodology are provided in Appendix A.2.

### 4.2 EXPERIMENTAL PROTOCOL

Across all Xayna methodologies, the core experimental unit involved presenting document pairs $(A, A')$ to the target LLMs in two primary orderings: $S_1 = (A, A')$ and $S_2 = (A', A)$. For certain methods like SPS-BPBE, more permutations or longer sequences with distractor documents were used. By comparing model behavior (outputs, internal representations, attention patterns) across these orderings, we isolate effects attributable to position from those attributable to the minimal content differences between $A$ and $A'$. We selected a representative set of widely-used LLMs, as reported in our results (e.g., Table 1 and Table 2): OpenAI: GPT-4o, Google: Gemini 2.5 (specific version, e.g., Gemini 1.5 Pro/Flash, varied based on API access), Meta AI: Llama 3 (e.g., Llama 3 Instruct models), Anthropic: Claude 3.7 (specific version, e.g. Claude 3 Opus/Sonnet, varied), Mixtral 8×22B, and DeepSeek's R1 (e.g. DeepSeek-LLM 67B). Access to models was via their respective APIs (for closed models) or locally hosted instances (for open models). Temperature was set to 0.1, a very low value, to minimize variability across runs and reduce hallucinations, while remaining stochastic. This setting yields highly consistent outputs in practice, unless stochasticity was integral to the method (e.g., DOPA with reasoning). Specific implementation choices for each Xayna method (e.g., layer for representation extraction, classifier type for CRLBD, specific attention heads for DPAF) are detailed within their respective method descriptions (Section 3), in the results (Section 5), or in Appendix A.3.

For closed-source, API-based models, full access to internal states such as representations, attentions, and gradients is not available. To address this limitation, we employed surrogate measures and carefully designed prompting strategies. For example, for PIRA we analyze embeddings of elicited summaries of the target document when placed at different positions, designed to approximate internal states. For DPAF, we prompt the model to "explain which parts of document A were most relevant for understanding document A', and vice-versa," to infer attentional focus. While other surrogate strategies could be imagined, our goal is to evaluate LLMs in a way that mirrors realistic user interaction. Prompting was therefore chosen as the primary surrogate, since it best imitates how end users naturally query and reason with these models. These are acknowledged as approximations, with full internal access used for open-source models where feasible (details in Appendix A.3). We anticipate that future users of Xayna, with deeper access to proprietary or research models, will be able to apply the framework more directly to hidden mechanisms.

## 5 EXPERIMENTAL RESULTS

Our empirical investigation, employing the Xayna framework across a suite of leading LLMs (GPT-4o, Gemini 2.5, Llama 3, Claude 3.7, Mixtral 8×22B, DeepSeek's R1), yielded consistent and compelling evidence of pervasive positional bias. This section details these findings, with quantitative support primarily drawn from Table 2 and other referenced figures/tables.

**Ubiquitous Nature of Positional Bias: An Overview** A striking overarching finding is that no model was immune to positional influences. Biases were often substantial, with merely reordering documents altering model outputs in a significant majority of cases (76-92% in prior DOPA-like reports). This pervasiveness suggests positional bias is a deeply ingrained characteristic. Representation-level (PIRA, CRLBD) and attention-based (DPAF) analyses revealed strong internal encoding of position, which translated to tangible output-level biases (DOPA, SPS-BPBE, PEA).

**Behavioral Manifestations: Preference and Choice (DOPA, SPS-BPBE)** DOPA experiments (Figure 2) revealed notable preference patterns based on order. SPS-BPBE provided robust Bayesian quantification, highlighting strong primacy effects in Table 2: Llama 3 showed a mean Pos1 preference of 0.892, and Claude 3.7 reached 0.795. Other models like Gemini 2.5 (0.603), Mixtral 8x22B (0.615), and DeepSeek's R1 (0.615) also preferred Position 1. GPT-4o (0.434) showed less pronounced primacy in this aggregate metric, suggesting more complex or balanced effects in this specific task. These values, often significantly above 0.5, indicate a statistically significant bias towards earlier information.

**Representational Geometry Distortion (PIRA, CRLBD)** PIRA results indicate that position fundamentally alters internal content representation. Figure 3(a) illustrates that CKA scores comparing representations of identical content at different positions are often low, signifying substantial structural dissimilarity and supporting the notion of "representational warping." CRLBD experiments (Figure 3(b)) corroborated this by showing that a classifier can predict a document's original position from its representation with high accuracy, implying strong and separable positional encoding.

**Attention Flow Asymmetries (DPAF)** DPAF analyses (Table 1 and "DPAF Asymmetry" in Table 2) revealed systematic asymmetries. Consistently, attention flow from the second document to the first ($F_{P2 \to P1}$) was greater than vice-versa ($F_{P1 \to P2}$). For instance, with Llama 3, $F_{P2 \to P1}$ was approximately 9.5% greater than $F_{P1 \to P2}$. This indicates a consistent pattern where earlier content acts as a more significant attentional anchor.

**Impact on Output Distributions and Confidence (PEA, TPBD)** PEA results (Table 2, $\Delta H$) showed that positional changes affect output uncertainty, with Mixtral 8x22B (0.106) exhibiting the highest average change. TPBD (KL divergence in Table 2) demonstrated that predicted token probability distributions are substantially altered by positional changes, with Llama 3 (2.292), Mixtral 8x22B (2.096), and GPT-4o (2.074) showing large divergences.

**Sensitivity to Positional Embeddings and Spectral Properties (PGS, FPBA)** PGS results ("PGS Gradient Difference" in Table 2, a ratio) indicated that models like Mixtral 8x22B (1.643) and GPT-4o (1.490) have outputs significantly more sensitive to earlier positional signals. FPBA ("FPBA Frequency Bias") showed that the spectral properties of representations are also affected by position, with GPT-4o (0.265) exhibiting the highest bias.

**Causal Impact and Inter-Document Influence (CPIA, LSP)** CPIA ("CPIA Relative Difference" in Table 2) quantified a consistently positive, albeit generally small, direct causal effect of position on outcomes (DeepSeek's R1: 0.0199; GPT-4o: 0.0189). LSP (conceptualized in Figure 4a) findings indicated that the processing of a target document $D_T$ is significantly influenced by the mere position of an accompanying document $D_P$.

**Model-Specific Patterns** Table 2 highlights distinct bias "fingerprints": **Llama 3** showed strong primacy (SPS-BPBE: 0.892) and high output token distribution sensitivity (TPBD KL: 2.292). **Mixtral 8x22B** exhibited the largest output entropy change (PEA $\Delta\mathcal{E}$: 0.106) and high gradient sensitivity (PGS: 1.643). **GPT-4o**, while showing nuanced choice behavior in this aggregate SPS-BPBE task (0.434), had high TPBD KL divergence (2.074) and the highest FPBA frequency bias (0.265). **Gemini 2.5** often showed comparatively lower bias scores (e.g., TPBD KL: 1.295), but still demonstrated clear positional effects (SPS-BPBE: 0.603). These variations underscore the need for model-specific audits.

**Inter-Metric Correlations** The Pearson correlation coefficients between Xayna's primary metrics, averaged across models (Figure 4b), were generally below 0.5. This underscores that the metrics capture relatively distinct aspects of positional bias, validating the multi-faceted nature of the Xayna framework. Despite this, some notable positive correlations emerged, suggesting underlying connections: PGS with PIRA-CKA ( 0.41); PEA $\Delta\mathcal{E}$ with TPBD KL Divergence ( 0.38); DPAF Asymmetry with CPIA Relative Difference ( 0.36); and LSP impact with TPBD KL Divergence ( 0.33). These relationships hint at links between embedding sensitivity, representational geometry, attention dynamics, and final output distributions.

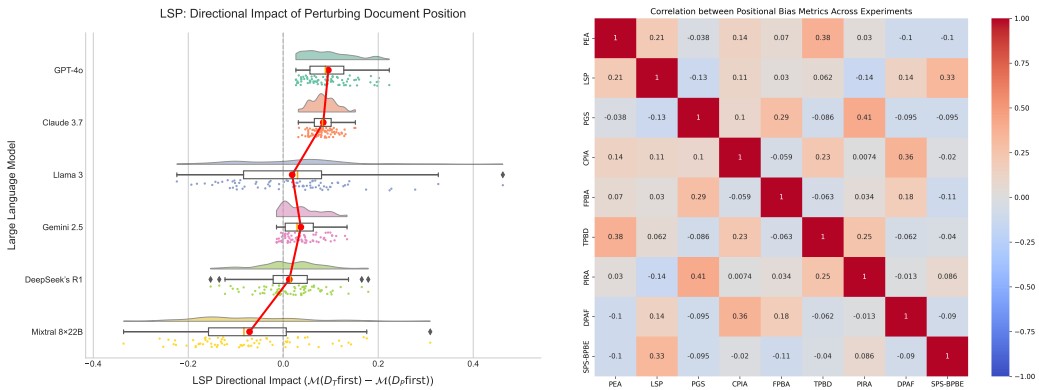

Figure 4: LSP conceptualization and inter-metric correlations for the Xayna framework. (a) Illustrates how LSP assesses the impact of a perturbing document's position on the interpretation of a target document. (b) Shows that Xayna's metrics capture distinct aspects of positional bias, with generally low to moderate correlations (diagonal values = 1; off-diagonals $\leq$ 0.5).

Table 2: Summary of positional bias metrics across models (mean $\pm$ sd). Higher values generally indicate stronger bias. Metric abbreviations: SPS-BPBE = primacy preference, PEA $\Delta\mathcal{E}$ = entropy change, TPBD = KL divergence, PGS = gradient sensitivity, FPBA = frequency bias, CPIA = causal impact, DPAF = attention asymmetry. (PIRA CKA scores are reported separately in Fig. 3a; lower CKA implies greater bias.)

| POSITIONAL BIAS METRIC | GPT-4o | GEMINI 2.5 | LLAMA 3 | CLAUDE 3.7 | MIXTRAL 8×22B | DEEPSEEK's R1 |
|---|---|---|---|---|---|---|
| SPS-BPBE Pos1 Preference | 0.434 (0.389) | 0.603 (0.589) | 0.892 (0.448) | 0.795 (0.452) | 0.615 (0.502) | 0.615 (0.436) |
| PEA $\Delta\mathcal{E}$ | 0.057 (0.092) | 0.060 (0.079) | 0.066 (0.081) | 0.068 (0.048) | 0.106 (0.095) | 0.073 (0.086) |
| TPBD KL Divergence | 2.074 (0.361) | 1.295 (0.254) | 2.292 (0.282) | 1.846 (0.445) | 2.096 (0.310) | 1.666 (0.336) |
| PGS Gradient Difference | 1.490 (0.030) | 1.453 (0.035) | 1.418 (0.035) | 1.439 (0.036) | 1.643 (0.033) | 1.397 (0.022) |
| FPBA Frequency Bias | 0.265 (0.197) | 0.126 (0.201) | 0.198 (0.245) | 0.145 (0.202) | 0.180 (0.176) | 0.199 (0.221) |
| CPIA Relative Difference | 0.0189 (0.0019) | 0.0069 (0.0021) | 0.0160 (0.0019) | 0.0165 (0.0020) | 0.0115 (0.0016) | 0.0199 (0.0014) |
| DPAF Asymmetry ($F_{P2\to P1} - F_{P1\to P2}$) | 0.020 (0.009) | 0.013 (0.012) | 0.024 (0.009) | 0.017 (0.009) | 0.022 (0.007) | 0.021 (0.009) |

## 6 DISCUSSION

The comprehensive empirical investigation facilitated by the Xayna framework establishes that positional bias is not a peripheral artifact but a fundamental characteristic of contemporary LLMs. These findings carry significant theoretical and practical implications.

The substantial dissimilarity of representations for identical content at different positions (PIRA), indicative of "representational warping," suggests that positional information fundamentally alters the embedding space, challenging notions of achieving true semantic abstraction independent of presentation order. This likely stems from how positional embeddings are integrated and processed. The consistent primacy effects (SPS-BPBE) and the attentional preference where later content directs more attention to earlier content (DPAF findings, see Table 1) echo cognitive phenomena like human serial position effects (Murdock Jr, 1968) and point towards an inductive bias of the transformer architecture itself. The observed discrepancies between strong internal biases (e.g., PIRA, DPAF) and sometimes more moderate output-level biases (e.g., PEA for some models in Table 2) suggest that LLMs may possess internal compensatory mechanisms, though these are clearly imperfect and

task-dependent. The relatively low inter-metric correlations (Figure 4b) further confirm that positional bias is not monolithic but a complex interplay of factors.

The prevalence of positional bias has immediate consequences for LLM applications. In multi-document summarization or question-answering, information presented earlier might be unduly favored (due to primacy effects like those quantified by SPS-BPBE in Table 2), undermining reliability and fairness. Prompt engineering strategies, such as permuting inputs, should be considered, and model auditing using frameworks like Xayna is crucial for sensitive tasks. Mitigation strategies can target different levels: **Data Augmentation** with permuted inputs; **Architectural Modifications** exploring novel PEs or more equitable attention mechanisms; **Regularization and Fine-tuning**, potentially using Xayna metrics like PIRA dissimilarity or CRLBD-style contrastive losses as part of the training objective to encourage positional invariance; and **Post-hoc Calibration** of outputs (Zhao et al., 2021). Future research should deepen the understanding of causal mechanisms and explore creating models that are position-aware (as order is sometimes functionally important) but not undesirably position-*biased*.

## 7 LIMITATIONS

While this study provides a comprehensive dissection using the Xayna framework, certain limitations should be acknowledged. First, the set of LLMs evaluated, though diverse and representative, is not exhaustive, and new architectures may exhibit different bias characteristics. Second, the dataset, while carefully constructed with programmatically generated document pairs (average length 5325 words) to control for content, might not fully replicate the nuances of naturally occurring multi-document scenarios or bias manifestations across vastly different context lengths or input granularities. Third, for commercial, closed-source models, the reliance on surrogate measures for internal states (e.g., for PIRA, DPAF as detailed in Appendix A.3) is an approximation compared to direct access available for open-source models. Fourth, Xayna's eleven methodologies, while extensive, may not capture every subtle form or manifestation of positional effects. Finally, the inter-metric correlation analysis provides initial insights into relationships but does not establish deeper causal links between the various observational metrics. Addressing these limitations offers important avenues for future research.

## 8 CONCLUSION

This paper introduced Xayna, a theoretically grounded and empirically robust framework for multi-faceted analysis of positional bias in large-language models. By integrating eleven complementary methodologies, Xayna goes beyond isolated observations to systematically dissect how the position of information influences the processing of LLM, from internal states to overt behavior. Experiments with state-of-the-art models (Section 5, Table 2) demonstrate that positional bias is significant and systemic. Key findings include: (1) LLMs form profoundly different internal representations for identical content based solely on position (PIRA, CRLBD); (2) attention mechanisms consistently privilege earlier information (DPAF); and (3) models display strong behavioral primacy effects (SPS-BPBE, e.g., Llama 3 and Claude 3.7). The scale of these effects, such as large KL divergences in output distributions (TPBD) and marked entropy shifts (PEA) under reordering, underscores the challenge they pose. Low correlations between metrics further show that Xayna captures diverse and complementary aspects of this phenomenon.

Xayna serves both as a diagnostic toolkit and a conceptual lens, revealing positional bias as a pervasive architectural property. Its insights highlight the urgent need for position-aware evaluation protocols and motivate principled mitigation strategies. As LLMs are increasingly deployed in decision-making contexts, addressing positional bias is critical to ensuring reliability, fairness, and genuine content understanding. While Xayna is not itself a mitigation technique, it provides a diagnostic lens that can complement existing strategies, such as data augmentation, regularization, or architectural modifications, by enabling more informed decisions about where and how to intervene. Exploring such integrations will be an important direction for future work, with the goal of reducing positional bias while preserving model performance.

By providing actionable metrics and a holistic diagnostic foundation, Xayna takes a key step toward building more robust and trustworthy language technologies.

## 9 ETHICS STATEMENT

This work investigates positional bias in large language models (LLMs) through controlled experiments on synthetic and publicly accessible data. No human subjects, personal data, or sensitive content were used. While the findings expose systematic weaknesses in LLMs, we do not release harmful prompts, adversarial datasets, or model weights. Instead, Xayna is designed as a diagnostic framework to support transparency, fairness, and responsible deployment of language technologies. We acknowledge that closed-source models limit access to internal states, and we emphasize that our surrogate probing methods are approximations. Future users with deeper model access should apply Xayna responsibly and in compliance with applicable terms of service for proprietary APIs. Details of LLM usage are provided in Appendix C.

## 10 REPRODUCIBILITY STATEMENT

We have taken several steps to ensure reproducibility. The eleven methodologies in Xayna are formally defined in Section 3, with mathematical formulations and pseudo-code provided in Appendix A.3. Dataset generation is described in Section 4, with additional implementation details in Appendix A.2. Hyperparameters, evaluation protocols, and model configurations are documented throughout the method and experimental sections, with summary tables included (e.g., Table 2). For open-source models, experiments can be fully reproduced; for closed-source models, we provide surrogate strategies that approximate internal analyses and can be replicated via API access.

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

# A  SUPPLEMENTARY MATERIAL

This section contains supplementary materials, including detailed algorithm parameters, further discussion of dataset generation, notes on implementation for specific Xayna methods, and pseudo-code for the Xayna methodologies.

## A.1  SUMMARY OF DEFINITIONS AND NOTATIONS

Table 3: Abbreviations of notions introduced in this paper.

| Abbrev. | Definition |
|---|---|
| DOPA | Measures order-driven choice bias by testing which of two semantically equivalent but reordered documents the model prefers. |
| SPS–BPBE | Estimates primacy/recency by shuffling positions across trials and inferring the probability of selecting position 1 with a Beta–Bernoulli model. |
| PEA | Quantifies how input reordering changes output uncertainty via the absolute difference in predictive entropy. |
| TPBD | Detects order effects on generation by comparing token probability distributions across orderings. |
| PIRA | Tests representation invariance by comparing internal representations of identical content placed at different positions. |
| CRLBD | Assesses whether position is encoded in hidden states by training a classifier to predict original position. |
| FPBA | Probes order sensitivity by analyzing frequency-domain signatures of token representations across positions. |
| DPAF | Quantifies directional attention flow between segments and measures asymmetry between later-to-earlier vs. earlier-to-later flows. |
| PGS | Measures positional sensitivity by comparing gradient norms with respect to positional embeddings across positions. |
| CPIA | Estimates the causal effect of content position on outcomes via interventions on positional embeddings or attention. |
| LSP | Evaluates how moving a perturbing document affects a target document's task performance for the same query. |
| PPS | Summarizes DOPA by reporting the fraction of trials that favor a given position, with deviation from 0.5 indicating bias. |

## A.2  FURTHER DATASET GENERATION DETAILS

The generation of document pairs $(A, A')$ involved specific prompting strategies for GPT-4o-mini to ensure semantic equivalence while introducing lexical and minor syntactic diversity. For each base document $A$, $A'$ was created with instructions to:

- Retain all core facts, arguments, and overall narrative structure.

- Substitute approximately 10-15% of words with synonyms.

- Rephrase a similar percentage of sentences or clauses without altering meaning.

- Avoid introducing new information or contradicting existing information.

- Distribute changes throughout the document rather than concentrating them.

The semantic similarity score of $0.92 \pm 0.03$ (using sentence-BERT embeddings) was monitored during generation to ensure consistency.

## A.3 XAYNA METHOD IMPLEMENTATION NOTES

**PIRA for API Models:** For models accessed via API where direct hidden states are unavailable, PIRA was approximated by prompting the model to generate a concise, abstractive summary of each document ($A$ and $A'$) when presented in sequences $S_1 = (A, A')$ and $S_2 = (A', A)$. The embeddings of these summaries (e.g., from a standard sentence encoder) were then compared using CKA. For example, if $A$ is at position 1 in $S_1$, the prompt would be: "Given Document 1: [Content of A] and Document 2: [Content of A'], provide a brief summary of Document 1." The same for $A$ at position 2 in $S_2$.

**DPAF for API Models:** For API models, DPAF's attention flow was approximated by prompting. After processing $S = (A, A')$, the model was asked, e.g., "When considering Document 2, which specific parts or sentences from Document 1 were most influential or relevant for your understanding of Document 2? Explain briefly." And vice-versa. The textual responses were analyzed qualitatively and, where possible, mapped back to segments to estimate attention focus. This is a heuristic approximation of direct attention weight analysis.

**CRLBD Classifier:** For CRLBD, a simple linear classifier (e.g., Logistic Regression) was typically trained on the representations. The goal is to assess linear separability of positional information. Using more complex classifiers could reveal non-linear positional encoding but makes the "bias" interpretation less direct.

**PGS Implementation:** PGS was typically implemented by taking the gradient of a chosen loss function (e.g., cross-entropy for a generation task, or a specific choice probability for DOPA-like tasks) with respect to the positional embeddings of the tokens belonging to the documents in question. The "PGS Gradient Difference" in Table 2 for this paper was calculated as the ratio of the mean L2 norm of gradients for tokens in the first document (semantically equivalent content) to that of the second document (semantically equivalent content in a different position). A value > 1 suggests higher sensitivity to the positional signal of the first document's content.

**CPIA Interventions:** For open-source models, interventions involved directly modifying positional embedding values (e.g., swapping PE vectors for two documents, or neutralizing PEs for a segment) or ablating specific attention heads hypothesized to contribute to positional effects. For API models, interventions were approximated by instructing the model to "ignore the position of Document X and treat it as if it were in position Y" or "focus solely on content from Document X, irrespective of Document Y." ATE was estimated by comparing outcomes (e.g., choice probability, answer correctness) across these interventional and control conditions.

---

**Algorithm 1** Document Order Preference Analysis (DOPA)

---

1: **Input:** LLM $f$, document pair $(A, A')$, number of trials $M$, task $T$ (e.g., "select more informative")
2: Initialize counts $N_1 \leftarrow 0, N_2 \leftarrow 0$ (preference for pos 1, pos 2)
3: **for** $m = 1 \rightarrow M$ **do**
4:     Create sequences $S_1 = (A, A'), S_2 = (A', A)$
5:     Get choice $C(S_1) = f(S_1, T)$ (model selects position 1 or 2 from $S_1$)
6:     **if** $C(S_1) = 1$ **then** $N_1 \leftarrow N_1 + 1$
7:     **else** $N_2 \leftarrow N_2 + 1$
8:     Get choice $C(S_2) = f(S_2, T)$ (model selects position 1 or 2 from $S_2$)
9:     **if** $C(S_2) = 2$ **then** $N_1 \leftarrow N_1 + 1$
10:     **else** $N_2 \leftarrow N_2 + 1$
                                        ▷ Adjusting for $A$ being effectively at pos 1
11: PPS$(1) = N_1/(2M)$
12: PPS$(2) = N_2/(2M)$
13: **Output:** PPS$(1)$, PPS$(2)$

---

**Algorithm 2** Permutation-Invariant Representation Analysis (PIRA)

1: **Input:** LLM $f$, document pair $(A, A')$, representation extraction function $h(\cdot)$
2: Create sequences $S_1 = (A, A')$, $S_2 = (A', A)$
3: Extract token representations for document $A$ in position 1: $H_A^{(1)} = h(A, \text{pos} = 1, S_1)$
4: Extract token representations for document $A$ in position 2: $H_A^{(2)} = h(A, \text{pos} = 2, S_2)$
5: Compute CKA score: $\text{CKA}(H_A^{(1)}, H_A^{(2)})$
6: (Alternatively, compute $1 - \text{cosine\_similarity}(\text{mean\_pool}(H_A^{(1)}), \text{mean\_pool}(H_A^{(2)}))$)
7: **Output:** CKA score (or other similarity/dissimilarity metric)

**Algorithm 3** Contrastive Representation Learning for Bias Detection (CRLBD)

1: **Input:** LLM $f$, set of documents $\mathcal{D}_{set}$, representation function $h(\cdot)$
2: Initialize dataset $\mathcal{X} = []$, $\mathcal{Y} = []$
3: **for** each document $D_i \in \mathcal{D}_{set}$ **do**
4:     For various positions $p_j$ and contexts $S_k$
5:     $H_{D_i}^{(p_j, S_k)} = h(D_i, \text{pos} = p_j, S_k)$
6:     Append $\text{mean\_pool}(H_{D_i}^{(p_j, S_k)})$ to $\mathcal{X}$
7:     Append label $p_j$ to $\mathcal{Y}$
8: Train classifier $g : \mathcal{X} \to \mathcal{Y}$ (e.g., Logistic Regression)
9: Evaluate accuracy of $g$ on a held-out test set
10: **Output:** Classifier accuracy

**Algorithm 4** Dynamic Positional Attention Flow (DPAF)

1: **Input:** LLM $f$ (with access to attention weights $A^{(l,h)}$), document pair $(A, A')$
2: Create sequence $S = (A, A')$
3: Identify token spans $T_A$ for $A$, $T_{A'}$ for $A'$
4: $\text{Flow}(A \to A') \leftarrow 0$, $\text{Flow}(A' \to A) \leftarrow 0$
5: **for** each layer $l$, each head $h$ **do**
6:     (Optional: get head/layer weight $w_{l,h}$)
7:     SumAttention $A \to A' \leftarrow \sum_{j \in T_A} \sum_{k \in T_{A'}} A_{jk}^{(l,h)}$
8:     SumAttention $A' \to A \leftarrow \sum_{j \in T_{A'}} \sum_{k \in T_A} A_{jk}^{(l,h)}$
9:     $\text{Flow}(A \to A') \leftarrow \text{Flow}(A \to A') + w_{l,h} \cdot (\text{SumAttention } A \to A'/|T_A|)$
10:     $\text{Flow}(A' \to A) \leftarrow \text{Flow}(A' \to A) + w_{l,h} \cdot (\text{SumAttention } A' \to A/|T_{A'}|)$
11: DPAF Asymmetry $= \text{Flow}(A' \to A) - \text{Flow}(A \to A')$
12: **Output:** $\text{Flow}(A \to A')$, $\text{Flow}(A' \to A)$, DPAF Asymmetry

**Algorithm 5** Positional Gradient Sensitivity (PGS)

1: **Input:** LLM $f$ (differentiable), document pair $(A, A')$, loss $L$
2: Create sequences $S_1 = (A, A')$, $S_2 = (A', A)$
3: For $S_1$: Identify positional embeddings $e_p(i)$ for tokens $i$ of $A$ (at pos 1). Compute $G_A^{(1)} = ||\nabla_{e_p(i)} L||$.
4: For $S_2$: Identify positional embeddings $e_p(j)$ for tokens $j$ of $A$ (at pos 2). Compute $G_A^{(2)} = ||\nabla_{e_p(j)} L||$.
5: Calculate PGS Metric, e.g., $|\text{mean}(G_A^{(1)}) - \text{mean}(G_A^{(2)})|$ or $\text{mean}(G_A^{(1)})/\text{mean}(G_A^{(2)})$.
6: **Output:** PGS Metric

---

**Algorithm 6** Frequency-Domain Positional Bias Analysis (FPBA)

---

1: **Input:** LLM $f$, document pair $(A, A')$, representation function $h(\cdot)$
2: $S_1 = (A, A')$, $S_2 = (A', A)$
3: $H_A^{(1)} = h(A, \text{pos} = 1, S_1)$ (matrix of token representations for $A$)
4: $H_A^{(2)} = h(A, \text{pos} = 2, S_2)$ (matrix of token representations for $A$)
5: Compute $\text{DFT}(H_A^{(1)})$ and $\text{DFT}(H_A^{(2)})$.
6: Analyze differences, e.g., $\sum_k |\text{Mag}(\text{DFT}(H_A^{(1)})[k]) - \text{Mag}(\text{DFT}(H_A^{(2)})[k])|$.
7: **Output:** Frequency-domain difference metric

---

# B  METRICS AND THEIR INTERPRETATIONS

For the definitions below, we consider a single pair of documents $(A, A')$ from our dataset, indexed by $i$. We form two sequences: $S_1 = (A_i, A'_i)$ and $S_2 = (A'_i, A_i)$. The model is denoted by $f(\cdot)$, its output probability distribution by $P(Y|\cdot)$, and its hidden state representation function by $h(\cdot)$. Each metric is typically calculated for every pair $i$ and then aggregated (e.g., by averaging) across the entire dataset.

## B.1  POSITIONAL ENTROPY ANALYSIS (PEA)

**Definition.** For each pair $i$, let $S_1 = (A_i, A'_i)$ and $S_2 = (A'_i, A_i)$. Compute the output entropy under both orders:

$$\mathcal{E}(Y|S_1) = -\sum_{t,v} P(y_t = v|S_1) \log_2 P(y_t = v|S_1),$$

$$\mathcal{E}(Y|S_2) = -\sum_{t,v} P(y_t = v|S_2) \log_2 P(y_t = v|S_2).$$

Define the per-pair entropy gap

$$\Delta\mathcal{E}_i = \big| \mathcal{E}(Y|S_1) - \mathcal{E}(Y|S_2) \big|, \quad \text{and the corpus statistic} \quad \text{PEB} = \frac{1}{N} \sum_{i=1}^{N} \Delta\mathcal{E}_i.$$

**Interpretation.** $\Delta\mathcal{E}_i \approx 0$ indicates order-invariant output uncertainty; larger values indicate order-sensitive confidence/uncertainty. High PEB implies systemic order sensitivity.

## B.2  PERMUTATION-INVARIANT REPRESENTATION ANALYSIS (PIRA)

**Definition.** For a document pair $(A_i, A'_i)$, create sequences $S_1 = (A_i, A'_i)$ and $S_2 = (A'_i, A_i)$. Extract the representations for document $A_i$ from both sequences: $H_A^{(1)} = h(A_i, 1, S_1)$ and $H_A^{(2)} = h(A_i, 2, S_2)$. Define the permutation-invariance violation using a dissimilarity metric like CKA or cosine distance on the mean-pooled vectors:

$$\text{PIV}_i = 1 - \text{CKA}(H_A^{(1)}, H_A^{(2)}),$$

or cosine gap $\delta_i = 1 - \cos\big(\text{mean\_pool}(H_A^{(1)}), \text{mean\_pool}(H_A^{(2)})\big)$. **Interpretation.** Values near 0 indicate order-invariant representations; larger $\text{PIV}_i/\delta_i$ indicate the representation itself encodes order.

## B.3  CONTRASTIVE REPRESENTATION LEARNING FOR BIAS DETECTION (CRLBD)

**Definition.** Build a probe dataset where the features are document representations and the labels are their original positions. For each document $D_i$ in a set, create multiple instances by placing it at different positions $p_j$ in various context sequences $S_k$:

$$\mathcal{Z} = \big\{(\text{mean\_pool}(h(D_i, p_j, S_k)), p_j)\big\}_{i,j,k}.$$

Train a linear classifier $\hat{y} = g(z)$ (or compute AUC with a linear score). Report:

$$\text{Acc}_{\text{probe}}, \ \text{AUC}_{\text{probe}}.$$

**Interpretation.** High linear separability (accuracy/AUC) indicates that *position* is easily decodable from the document's representation, i.e., the representation carries positional information. Chance-level performance suggests near order-invariance.

### B.4 DYNAMIC POSITIONAL ATTENTION FLOW (DPAF)

**Definition.** Let $\alpha_{ij}$ denote attention from source token $i$ to target token $j$. Define cross-document flows

$$\mathrm{PAF}(A \to B)_i \;=\; \frac{1}{|A_i||B_i|} \sum_{u \in A_i} \sum_{v \in B_i} \alpha_{uv}, \qquad \mathrm{PAF}(B \to A)_i \;=\; \frac{1}{|B_i||A_i|} \sum_{u \in B_i} \sum_{v \in A_i} \alpha_{uv}.$$

Define asymmetry

$$\Delta_i^{\mathrm{PAF}} \;=\; \big| \mathrm{PAF}(A \to B)_i - \mathrm{PAF}(B \to A)_i \big|.$$

**Interpretation.** Larger $\Delta_i^{\mathrm{PAF}}$ indicates directional attention imbalance attributable to ordering; near-zero suggests symmetric cross-document integration.

### B.5 STOCHASTIC PROMPT SHUFFLING & BAYESIAN POSITIONAL BIAS (SPS–BPBE)

**Definition.** For each pair $i$, run $K$ randomized-order trials yielding binary choices $Y_{ik} \in \{0, 1\}$ indicating selection of a designated reference document (e.g., $A_i$). Partition counts by whether $A_i$ was shown first ($F$) or second ($S$):

$$x_F = \sum_{k \in F} \not\Vdash\{Y_{ik} = 1\}, \quad m_F = |F|; \qquad x_S = \sum_{k \in S} \not\Vdash\{Y_{ik} = 1\}, \quad m_S = |S|.$$

Use Beta posteriors with prior $\mathrm{Beta}(\alpha_0, \beta_0)$:

$$\theta_F \sim \mathrm{Beta}(\alpha_0 + x_F, \; \beta_0 + m_F - x_F), \qquad \theta_S \sim \mathrm{Beta}(\alpha_0 + x_S, \; \beta_0 + m_S - x_S).$$

Define a bias index $\beta_i = \mathbb{E}\big[|\theta_F - \theta_S|\big]$ (approximated by Monte Carlo) and report credible intervals for $\theta_F - \theta_S$. **Interpretation.** $\beta_i \approx 0$ (with a CI containing 0) indicates no detectable order effect; larger $\beta_i$ and CIs excluding 0 indicate robust positional bias for pair $i$. Aggregate $\overline{\beta} = \frac{1}{N} \sum_i \beta_i$ for corpus-level sensitivity.

## C LLM USAGE

Large Language Models (LLMs) were used only as general-purpose assistive tools for minor grammar correction and language polishing. They were not involved in research ideation, technical content creation, analyses, or experiments. All technical contributions and results were authored and verified solely by the research team.

