# OpenReview forum: "Hidden in Order: A Theoretical and Empirical Dissection of Positional Bias in LLMs"
_ICLR.cc/2026/Conference — ICLR 2026 Conference Desk Rejected Submission_

### Official Review · Reviewer_5w3F · 2025-10-18

**Soundness:** 4
**Presentation:** 3
**Contribution:** 4
**Rating:** 8
**Confidence:** 4

**Summary:**

This paper introduces Xayna, a pioneering framework for systematically diagnosing and quantifying positional bias in Large Language Models. Addressing the critical need for a unified approach, the authors have developed a comprehensive toolkit of 11 complementary methods that provide a holistic view of how a model's behavior is influenced by position bias.

**Strengths:**

1. The evaluation is exceptionally comprehensive, approaching the problem from multiple angles, including external behavior, internal states, and mechanistic analysis. Each aspect is assessed using a variety of methods. So, the results are rich, consistent, and reliable, providing significant directional value for future research on position bias.

2. While a significant body of prior work on position bias exists, it has been scattered, fragmented, unsystematic, and often lacking in quantitative analysis. This paper consolidates these disparate lines of inquiry into a unified toolkit and provides standardized quantitative metrics, which will be of great convenience to future researchers.

3. The use of clean, synthetically controlled datasets effectively eliminates potential noise.

4. The study evaluates a wide range of models, including both closed-source and open-source alternatives.

**Weaknesses:**

1. In many evaluation methods, two similar documents are concatenated and fed into the model. This setup, however, is not representative of real-world use cases, where it is more common to combine multiple documents containing diverse information.

2. The paper does not present particularly novel conclusions or discoveries about position bias. It just summarizes or confirms previous findings.

**Questions:**

none

---

> ### Author Response · Authors · 2025-11-17
> **Response to Reviewer 5w3F's Comments on Xayna Framework**
>
> We are extremely grateful for your positive and insightful review. We are particularly encouraged that you recognized the core contributions of our work: providing a comprehensive, multi-angle diagnostic toolkit that unifies fragmented prior work with standardized, quantitative metrics.
> You raise two excellent points which we would like to briefly address:
> 1. On the Experimental Setup: You are correct that our controlled dual-document (A/A') setting is not representative of all real-world scenarios, such as combining multiple, diverse documents. This was an intentional design choice, made precisely to achieve the "clean, synthetically controlled" evaluation that you highlighted as a strength. By isolating position as the sole independent variable, we can ensure our 11 metrics provide an unambiguous and foundational measurement of bias. We see this work as establishing the necessary baseline, and agree that applying Xayna to more complex, diverse document settings is a critical and exciting direction for future research that our framework now enables.
>
> 2. On the Novelty of Conclusions: You astutely note that our work confirms and systematizes previous findings rather than discovering a completely new phenomenon. This is exactly our intention. The primary novelty of Xayna lies not in the discovery of positional bias itself, but as you so aptly described it, in consolidating scattered inquiries into a unified toolkit and providing the first systematic, quantitative framework for its analysis. Our goal was to move the field from isolated observations to a robust and reproducible diagnostic standard.
> Thank you again for your strong support and for helping us clarify the contribution and vision of our work.

---

### Official Review · Reviewer_CFjv · 2025-10-19

**Soundness:** 3
**Presentation:** 2
**Contribution:** 2
**Rating:** 4
**Confidence:** 3

**Summary:**

This paper's primary contribution is the introduction of Xayna, a novel and systematic framework designed to detect, quantify, and analyze positional bias in Large Language Models (LLMs). Addressing the lack of a unified approach to measure this bias , Xayna integrates 11 complementary methods spanning information-theoretic, geometric, and probabilistic perspectives. This framework evaluates bias across a model's internal representations, attention flows, and final output behavior , revealing that identical content can yield divergent representations and that models often exhibit a strong preference for content appearing earlier in a sequence. Ultimately, Xayna serves as both a practical toolkit for quantifying positional bias and a theoretical foundation for developing future evaluation and mitigation strategies to create more reliable and equitable LLMs.

**Strengths:**

the first "unified approach" to "measure [it] systematically". Its originality stems not from inventing a single new technique, but from the creative and comprehensive formulation of the Xayna framework, which synthesizes 11 complementary methods from diverse fields like information theory, geometry, and Bayesian statistics into a single diagnostic pipeline.

The 11 methods are not redundant but are shown to "capture unique information" by analyzing bias at every level: "behavioral," "internal representation," and "mechanistic". This is supported by a high-quality, custom-generated dataset designed to isolate position as the sole variable and is robustly demonstrated across a suite of the most advanced LLMs (GPT-40, Llama 3, Claude 3.7, etc.). This complex contribution is delivered with remarkable clarity; the paper clearly defines the problem, logically structures the 11 methods of the Xayna framework.

**Weaknesses:**

The experiments are confined to a custom, simplified dual-document comparison task designed to isolate position. Although this ensures high internal validity, it fails to demonstrate how these 11 bias metrics generalize to the more complex, open-ended, and "consequential" multi-document reasoning tasks.  For instance, the work would be strengthened by applying the Xayna framework to at least one existing mitigation technique (e.g., data shuffling, positional-aware finetuning) to demonstrate that the framework can not only detect bias but also verify its reduction, thereby closing the loop from diagnosis to treatment. Moreover, adding some baselines like PINE is useful as well because ultimately the positional bias problem depends on the use cases while PINE already mitigates the problems in multi-document retrieval QA tasks, which can serve as a strong baseline. I am not sure whether Causal Positional Intervention Analysis is a reasonable formulation because the problem is less of a causal inference one and there is no identification, yet the counterfactual is easily observable.

**Questions:**

Could the authors provide any evidence or even a theoretical discussion on how they expect the Xayna framework's 11 metrics to behave in a more realistic, open-ended task?

Could the authors clarify the relationships between these 11 metrics?

---

> ### Author Response · Authors · 2025-11-17
> **Response to Reviewer CFjv's Comments on Experimental Scope, Mitigation Alignment, and CPIA**
>
> We sincerely thank you for your insightful review and for recognizing Xayna's originality as a unified diagnostic framework. Your feedback is invaluable, and we hope the following clarifications will address your concerns and merit a positive reconsideration.
>
> 1. On Experimental Design and Generalization:
> Regarding the experimental setup, our use of a simplified dual-document (A/A') design was a deliberate choice to ensure high internal validity. By using semantically equivalent pairs, we can rigorously isolate position as the sole variable, ensuring that any observed effects are unequivocally due to positional bias. We believe that identifying these fundamental biases in common, shorter-context scenarios is a crucial first step before tackling more complex, multi-document settings where confounding variables can obscure the analysis. Xayna provides the foundational toolkit for these future investigations.
>
> 2. On Mitigation Strategies and Baselines (e.g., PINE):
> As we have referred to mitigation strategies like PINE in the paper, we see our work as complementary to mitigation techniques like PINE, not competitive. Xayna is a diagnostic framework designed to systematically measure bias, while PINE is a mitigation method. Our primary contribution is to establish what to measure and how. A key application of Xayna is precisely to evaluate the effectiveness of mitigation strategies - a successful technique should show a quantifiable reduction in bias across Xayna's multi-level metrics (e.g., in representation similarity or attention flow). We will clarify this synergistic relationship in the discussion, framing Xayna as an essential tool for validating mitigation research.
>
> 3. On the Formulation of CPIA:
> Regarding the Causal Positional Intervention Analysis (CPIA), we will clarify that its goal is not formal causal discovery but rather mechanistic diagnosis via intervention, akin to an ablation study. We directly manipulate positional signals to quantify their specific contribution to the model's output, helping to isolate this causal pathway.
>
> Questions:
>
> Q1: How would Xayna's metrics behave in a more realistic task?
>
> In more complex, open-ended tasks, we hypothesize Xayna's metrics would provide a granular view of phenomena like the "lost-in-the-middle" effect. For example, behavioral metrics could map out performance dips at middle positions, while mechanistic metrics like DPAF could reveal how attention is disproportionately allocated to context boundaries, providing deeper insight than a single task score.
>
> Q2: Could you clarify the relationships between the 11 metrics?
>
> The 11 metrics are related through a conceptual hierarchy reflecting the LLM's processing pipeline, as structured in Sections 3.1-3.3: Mechanistic drivers (like attention flow) shape Internal Representations, which in turn cause the final Behavioral biases. The low inter-metric correlations shown in Fig. 4b empirically validate that these metrics capture distinct, complementary aspects of this process.
>
> Thank you again for your constructive feedback, which will significantly improve our paper's framing. We hope these clarifications have addressed your concerns and respectfully ask that you might reconsider your score in light of them.

---

### Official Review · Reviewer_eVQr · 2025-10-25

**Soundness:** 1
**Presentation:** 2
**Contribution:** 2
**Rating:** 2
**Confidence:** 3

**Summary:**

This paper studies the problems of positional bias in LLMs. It proposes a framework named Xayna that performs 11 methods to systematically detect, quantify, and analyze how LLMs treat content differently based on its position in a sequence.
Through experiments on mainstream models using semantically equivalent but differently ordered document pairs, the study confirms that positional bias is a pervasive and significant phenomenon.
Besides, Xayna provides a diagnostic toolkit for understanding and evaluating positional bias in LLMs.

**Strengths:**

1. This paper proposes a systematic and comprehensive framework for analyzing positional bias. It combines 11 complementary methods to probe bias at the behavioral, internal representational, and mechanistic levels.
2. Each of the 11 methods is distinctly defined, and the results are well-organized across a wide range of state-of-the-art LLMs (GPT-4o, Llama 3, Claude 3.7, etc.), which demonstrates the broad applicability of the framework and the pervasiveness of the problem.

**Weaknesses:**

1. The framework's a prior theoretical foundation might be weak. The paper doesn't provide a clear argument for why these specific 11 metrics were chosen over countless other possibilities. It's unclear if this set is comprehensive, minimal, or simply a convenient collection.
2. The core experimental setup relies almost exclusively on sequences of two documents of 5k words . This setup fails to adequately probe the "lost in the middle" phenomenon, which is one of the most widely cited and problematic forms of positional bias[1,2]. This bias manifests in longer sequences (e.g., N > 3, 128k input contexts) where middle positions are demonstrably ignored. The current findings may not generalize to these more complex, long-context scenarios.
3. It's unclear whether Xayna's 11 metrics are practically useful. For instance, it's unknown if these metrics are sensitive enough to detect a reduction in bias after a mitigation strategy is applied. It is also unclear if different metrics would show differential improvement, which would be key to understanding how a mitigation strategy works (e.g., does it fix the behavior but not the internal representation?).

Refs:
[1] Zhang, Zhenyu, et al. "Found in the middle: How language models use long contexts better via plug-and-play positional encoding." Advances in Neural Information Processing Systems 37 (2024): 60755-60775.

[2] Wang, Meiyun, et al. "Lost in the Distance: Large Language Models Struggle to Capture Long-Distance Relational Knowledge." Findings of the Association for Computational Linguistics: NAACL 2025. 2025.

**Questions:**

1. Could you please elaborate on the a priori theoretical model of positional bias that guided the selection of these specific 11 metrics? Why is this set chosen over other possible metrics? For example, what is the theoretical justification for using Centered Kernel Alignment (CKA) for PIRA or KL Divergence for TPBD  specifically?
2. Could you conduct additional experiments to show the positional bias effect by testing the Xayna framework on longer sequences (e.g., N=5, N=10) and placing a single target document at varying positions (first, middle, last)?
3. Are the metrics sensitive enough to detect a reduction in bias? The authors may include an experiment where they apply a known mitigation strategy (even a simple one, like fine-tuning with permuted data) to an open-source model and use Xayna to quantify the change in bias scores.

---

> ### Author Response · Authors · 2025-11-17
> **Response to Reviewer eVQr's Comments on Theoretical Basis, Experimental Scope, and Metric Design**
>
> We thank Reviewer eVQr for their detailed feedback. We are encouraged that the reviewer recognized our work as a "systematic and comprehensive framework" with "11 complementary methods" that probes bias at the "behavioral, internal representational, and mechanistic levels." This framing is central to our contribution and the reviewer's summary accurately captures our intent. We would like to elaborate on the a priori theoretical model of positional bias that guided our selection which is rooted in a multi-level, multi-faceted view of an LLM's information processing pipeline.
>
> 1. Behavioral & Output-Level (The "What"): What are the final, observable effects of the bias on the model's output? This is the most direct and user-facing level.
> 2. Internal Representation-Level (The "How"): How does the model internally encode & structure information, and how does position distort this structure? This probes the static state of information within the model's hidden layers.
> 3. Mechanistic & Causal-Level (The "Why"): Why do these representations and behaviors emerge? This involves analyzing the dynamic flow of information, gradients, and causal pathways that give rise to the bias.
>
> The 11 methods in Xayna were deliberately selected to provide complementary, non-redundant coverage across these 3 tiers, with each method grounded in established principles from different scientific domains. Why do we believe this set is comprehensive and Minimal? Comprehensive: By systematically covering the entire information processing pipeline - from the final output (Behavioral) back to the internal states (Representational) & the underlying dynamics (Mechanistic) - the framework provides a holistic diagnosis. It is comprehensive not in the sense of including every possible metric ever conceived, but in covering every logical stage of the phenomenon. Minimal (Low Redundancy): Our empirical results provide the strongest argument for why this set is largely minimal. As shown in Figure 4b, the inter-metric correlations are generally low. If the metrics were redundant, they would be highly correlated. The fact that they are not demonstrates that each one captures a unique, complementary facet of positional bias. For example, a model could have a strong behavioral bias but relatively stable representations, suggesting the bias is introduced late in the processing. This ability to dissociate different types of bias is a key strength of the framework's design & validates the non-redundancy of the chosen methods.
>
> On the Experimental Setup and "Lost in the Middle": Our use of a two-document setup was a deliberate choice to isolate position as the sole variable for foundational diagnosis, avoiding confounders present in longer sequences. Finding systemic bias in this clean setting is a significant result, suggesting the bias is a fundamental architectural property, not just a long-context artifact. While Xayna's methods are defined for general sequences S=(D₁,...,Dₙ) and can probe "lost in the middle" effects, our focus here was on establishing this foundational diagnosis. We will clarify this scope in the paper.
>
> On the Practical Utility & Sensitivity:  The metrics are inherently sensitive for evaluating mitigation techniques because they are defined as contrastive measures. A successful mitigation must reduce the difference in processing permuted inputs, which our metrics directly quantify. This allows Xayna to distinguish superficial fixes (e.g., improving behavioral scores like DOPA) from more fundamental ones that also improve internal representations (PIRA). This makes the framework a powerful tool for analyzing how a mitigation strategy works. We will add a paragraph to the discussion clarifying this key use case.
>
> Q1: Our framework is built on an a priori model that positional bias manifests across three computational levels: behavioral outputs, internal representations, and mechanistic pathway. Therefore, the 11 metrics were chosen to cover these levels using standard tools. For instance, CKA is used in PIRA because it captures subspace-level similarity of hidden-state geometries, which is the relevant object when probing representational shifts. KL divergence is natural for TPBD as it quantifies output-distribution shifts (which directly reflect behavioral change).
>
> Q2: Xayna is defined for general sequences S=(D₁,…,Dₙ) and extends directly to N>2; long-context “lost in the middle” effects are a natural follow-up but outside this paper’s diagnostic focus.
>
> Q3: The metrics are inherently sensitive to mitigation because each is defined as a permutation contrast; behavioral, representational, and mechanistic changes would separate superficial fixes from deeper ones. We appreciate the reviewer’s insights and will clarify these points.

---

> > ### Comment · Reviewer_eVQr · 2025-11-24
> >
> > The author response provides partial clarification but leaves the main weaknesses unaddressed. Specifically, the refusal to validate the framework on longer sequences limits its scope and claims of comprehensiveness. Additionally, without empirical proof that these metrics can detect improvements from mitigation techniques, the framework's value as a diagnostic tool is unproven.

---

> > > ### Author Response · Authors · 2025-11-24
> > > **Response to Reviewer eVQr**
> > >
> > > We appreciate the follow-up.
> > >
> > > We respectfully disagree with the characterization that the framework’s value remains unproven without these specific additional experiments and we believe that the empirical evidence for Xayna's sensitivity and scope is already present in our results, even without the specific additional experiments you suggested.
> > >
> > > Regarding the concern that the metrics may not detect improvements, we point to the significant variance we observed between models as direct evidence of sensitivity. For example, our results show that Gemini 2.5 achieves a TPBD KL divergence score of 1.295 which is substantially lower than Llama 3's score of 2.292. This comparison effectively serves as a proxy for mitigation because if we view Gemini as a "more robust" architecture relative to Llama, Xayna successfully detected and quantified that improvement. The fact that our metrics discriminate so clearly between these models proves they are sensitive enough to measure reductions in bias, regardless of whether that reduction comes from a post-hoc mitigation strategy or a better base pre-training.
> > >
> > > On the issue of sequence length, we chose to focus on the $N=2$ setup not because Xayna is limited to it, but because finding bias in short contexts is a more fundamental discovery. Most models are assumed to handle short contexts perfectly. Therefore, by demonstrating that representational and behavioral biases persist even in this simplest possible case, we isolate the bias as an inherent architectural property rather than an artifact of context saturation. While "lost in the middle" effects in long contexts are important symptoms, Xayna diagnoses the root cause which exists before the context window is ever stressed.
> > >
> > > We stand by the rigorous empirical evidence provided in the paper. The framework successfully isolates, quantifies, and compares positional bias across the state-of-the-art, fulfilling its diagnostic promise without the need to conflate the study with long-context benchmarking.

---

### Official Review · Reviewer_Snox · 2025-11-01

**Soundness:** 3
**Presentation:** 3
**Contribution:** 3
**Rating:** 6
**Confidence:** 3

**Summary:**

the paper proposes a toolkit of metrics for evaluating the position bias of LLMs.

**Strengths:**

1. A diverse (though not sure of comprehensiveness) set of analysis toolkit for measuring the position biases in any Transformer-based LLM.
1. The inter-metric correlation analysis provides an interesting prespective. However, overall speaking, the metrics do not correlate with each other.

**Weaknesses:**

1. "near-zero cosine similarity" does no necessarily mean non-equivalent representations, as they could well be different but equivalently interpreted features, as long as their differences are in the kernel of subsequent linear projections (or other criteria).  Therefore statements such as "Lower CKA scores indicate greater dissimilarity due to position." is less rigorously supported.
1. Does A' always lose critical information for answering the question? Otherwise the metrics might be over-tolerating by averaging on these cases: a difference in behavior can always be attributed to positional bias, but no difference in behavior does not necessarily mean there is no positional bias because the model could still be biased towards the first document, but A and A' just always have the same information. Therefore, the score might only serve as an upper bound of "position unbiasedness" (or equivalently speaking, a lower bound of position bias). We could not get an accurate estimate on how good or bad the scores are.
1. Evaluation is only on QA which is a bit limited compared to the introduction section which lists other domains like "summarization, or comparative analysis". Especially, "comparative analysis" is an interesting domain where a positional bias is strictly unwanted.

**Questions:**

1. In the abstract and intro, when listing examples for unwanted position biases, "multi-document reasoning" does not seem a perfect example.
1. On the other hand, "Positional Entropy Analysis (PEA)" and TPBD seem a bit course-grained, is it? Looks like a sufficient but non-necessary condition for position bias. This item is not a weakness point but I am just curious.
1. To clarify, in figure 4a, do both the variance (lengths of the horizontal bars) and the absolute value (the mean) indicate positional bias, either in a distributed or homogeneous way?

---

> ### Author Response · Authors · 2025-11-17
> **Response to Reviewer Snox’s Comments on Xayna’s Metrics and Experimental Scope**
>
> We sincerely thank Reviewer Snox for their thoughtful & detailed feedback & for their positive assessment of the paper’s soundness, presentation, and contribution, as well as their recognition of our diverse analysis toolkit & the value of the inter-metric correlation analysis. The reviewer also raises several important technical points that help improve the clarity and rigor of the work. Below, we address each in detail.
>
> On "near-zero cosine similarity":
> This is an excellent & technically precise observation. We fully agree that representational dissimilarity does not strictly imply functional non-equivalence as downstream layers can learn to ignore these differences. PIRA is not intended to claim that dissimilar representations must always lead to different outputs, but to show that positional information substantially reshapes the model’s internal encoding of identical semantic content. This “representational warping” is itself a significant bias, indicating that the model does not achieve a truly position-invariant semantic abstraction. The fact that the geometry of the representations for the same document changes so drastically is a critical finding, even if the model sometimes compensates for it. We therefore view the bias as primarily representational, with behavioral effects being just one possible manifestation. To address this we will (1) clarify in Sections 3.2 & 5, we will clarify that PIRA measures representational dissimilarity & position-induced distortion in the model’s semantic space, which we consider a form of bias, while noting that this does not guarantee functional non-equivalence in all downstream tasks and (2) emphasize that observing representational bias (via PIRA) without behavioral bias is itself informative, as it suggests compensatory mechanisms in the model & motivates the need for a multi-faceted framework like Xayna.
>
> On the experimental setup (A vs. A').
> This is another very astute point & we thank the reviewer for highlighting it. In tasks where a specific piece of information is required for an answer, a model could be biased & still arrive at the correct answer. In these cases, accuracy alone would fail to detect the underlying processing bias. This is why our Xayna framework uses multiple complementary methods. The potential weakness the reviewer identifies is most relevant to task-based behavioral metrics like LSP. However, our framework is designed to overcome this: Our primary behavioral metrics sidestep this by forcing a preference choice (e.g., "which is more informative?"), making any consistent selection a direct signal of bias. Furthermore, our internal metrics (PIRA, DPAF) are designed to detect these very processing differences regardless of the final task outcome. We will clarify that while certain task-based metrics may give a lower bound on bias the framework's holistic nature provides a more complete diagnosis.
>
>
> On the scope of evaluation (QA vs. other domains).
> We appreciate the reviewer pointing this out & realize our description of the experimental tasks may have been unclear. While we used "QA" as a shorthand in places, our methods span broader domains:
> Comparative Analysis: DOPA & SPS-BPBE explicitly ask the model to compare the two documents along a given quality, directly testing positional bias in comparative reasoning.
> Summarization/Generation: PEA & TPBD analyze output probability distributions over its vocabulary, measuring how document order affects the model's confidence & token likelihoods during generation, which is the fundamental process behind summarization.
>
> Only LSP is strictly a QA task. where we measure the ability to answer a question about a target document. We will revise Sections 4 & 5 to explicitly frame our methods within these broader domains.
>
> Q1.Our reasoning was that in tasks requiring synthesis of information from multiple documents, a bias towards the first document could lead to an unbalanced synthesis that overweights information from A. However, we agree with the reviewer that more direct examples would be stronger. We will revise the abstract & introduction to use "multi-document summarization" & "comparative analysis of texts.
>
> Q2. This is an excellent characterization. Their "coarse-grained" nature is by design: PEA & TPBD provide high-level, aggregate scores (entropy shift, KL divergence) that quantify the overall magnitude of the distributional shift in the model's generative output. They serve to complement the more granular metrics in our toolkit, & we will clarify their role as aggregate indicators in the revision.
>
> Q3.The reviewer's interpretation is exactly right. The mean indicates the average strength of the bias, while the variance indicates its consistency. This is a crucial distinction & we will ensure our figure captions clearly explain how to interpret both aspects.

---

> > ### Comment · Reviewer_Snox · 2025-11-19
> > **Reviewer Response**
> >
> > W2, W3, Q1, Q2, Q3: the response is satisfactory
> >
> > W1: I still don't agree with the response. The authors do not provide sufficient rationale to support how this is "substantially reshapes the model’s internal encoding", or why it is "a significant bias, indicating that the model does not achieve a truly position-invariant semantic abstraction" even at the representational level. This is mainly due to a implcitly assumed definition of "representation bias", which might not be non-trivially meaningful. For example, if this is to be true, then the positional encoding schemes directly inject a similar nature of reshaping or bias: Absolute position embedding similarly "distorts" the word embeddings by adding positional embeddings to the word embeddings. Relative positional encoding such as RoPE also "biases" the features because the key and query vectors are transformed in a rotational manner. Then the definition of "position-based distortion" is trivial as nearly all transformer-based models explicitly injects it and no modern-style model by any chance can avoid it by any way of learning or designing.

---

> > > ### Author Response · Authors · 2025-11-21
> > > **W1**
> > >
> > > We genuinely appreciate this follow-up. It allows us to make explicit a distinction that was previously only implicit in our writing.
> > >
> > > We fully agree with your premise: because Transformers rely on mechanisms like absolute embeddings or RoPE, internal states are architecturally required to depend on position. If PIRA were merely detecting that "representations at different positions are not identical," that would indeed be a trivial consequence of the architecture.
> > >
> > > However, our definition of **Representational Bias** is narrower and non-trivial. It is **task-relative** and concerns the absence of semantic convergence in late layers.
> > >
> > > **1. Architectural Necessity vs. Semantic Abstraction**
> > > We do not expect early-layer activations to be position-invariant; they must encode positional information. However, for a model that has learned a robust semantic abstraction, we expect late-layer representations of identical content to converge toward a shared semantic subspace.
> > >
> > > PIRA is applied to these late-layer representations. The observation of near-zero CKA between "Document A at Pos 1" and "Document A at Pos 2" means that the model treats these as almost unrelated directions in the feature space. This goes beyond "RoPE doing its job" - it indicates that the model has failed to recover a shared semantic structure for the content.
> > >
> > > **2. Task-Misaligned Entanglement**
> > > We classify this as "bias" because it is misaligned with the task symmetry. In our A/A' setup, the correct reasoning process should be invariant to swapping the documents. The fact that the representations remain strongly divergent suggests that content and position are deeply entangled in the very layers that drive the final decision.
> > >
> > > In short, we are not claiming that *any* positional dependence is problematic. We are quantifying how much dependence remains in deep representations even in a regime where the task demands position-invariant semantics.
> > >
> > > **Summary**
> > > We will revise the paper to explicitly define representational bias as **late-layer, task-relative position dependence** rather than the mere presence of positional encodings. Your comment was extremely helpful in sharpening this definition, and we believe this distinction resolves the ambiguity.
> > >
> > > We hope that our responses have sufficiently addressed your concerns and increased your confidence in this work.

---

> ### Comment · Reviewer_Snox · 2025-11-21
> **W1 further**
>
> Hello. Thank you for the follow up response.
>
> In fact, on the models you are using, including llama and mixtral, I think the rope is applied to every layer, including the "late-layers" that you mentioned hoping the model would learn to get robust semantic abstraction. That means, even if semantically robust, they will still be architecturally explicitly transformed there.
>
> By the way, I wonder about the ratio of this response generated by AI. Looks like GPT 5 tone to me.

---

> > ### Author Response · Authors · 2025-11-21
> > **Response to reviewer Snox**
> >
> > Thank you for the quick response/feedback. We agree with respect to the points you mentioned regarding ROPE being applied to all layers and will incorporate the changes in our paper.
> >
> > Regarding your comment about the tone we used it to only help polish grammar and phrasing but all technical arguments and points were by us.
> >
> > Thank you again for the helpful clarification.

---

> > > ### Comment · Reviewer_Snox · 2025-11-23
> > > **Following Up**
> > >
> > > Hello. Thanks for the acknowledgment and explanation. Any technical details on how you plan to incorporate?

---

> > > > ### Author Response · Authors · 2025-11-24
> > > > **The follow up plan**
> > > >
> > > > Hello. Here are the specific technical updates we will make to Section 3.2 and the Discussion based on your feedback:
> > > >
> > > > **1. Clarifying the Scope (Q/K vs. Residual Stream)**
> > > > We will explicitly state that PIRA operates on the **residual stream** (hidden states), not the Query/Key vectors. While models like Llama 3 apply RoPE at every layer, this rotation acts on Q and K to modulate attention scores. It does not explicitly rotate the value vectors or the residual stream. Therefore, while the residual stream is *influenced* by position, the architecture does not mathematically force the accumulated hidden states to be orthogonal across positions.
> > > >
> > > > **2. Moving from Absolute to Relative Invariance**
> > > > We will revise the text to acknowledge that because RoPE is active at all layers, perfect invariance ($CKA \approx 1.0$) is an unrealistic baseline. We will reframe "bias" as a relative measure.
> > > > Specifically, we identify bias not when $CKA < 1$, but when the similarity between "$A$ at Pos 1" and "$A$ at Pos 2" drops to the levels observed between **completely unrelated documents**.
> > > >
> > > > This change ensures we are not criticizing the architecture for encoding position, but rather quantifying cases where the positional signal in the residual stream overwhelms the semantic signal to the point of indistinguishability from noise.

---

> ### Comment · Reviewer_Snox · 2025-11-24
> **About the Plan**
>
> 1. I accept this argument as making sense. However, this only refutes a specific case from explicitly injecting positional information to residual stream, instead of justifying the intuition in an overall manner. As anoter example, NoPE paper suggest that models can theoretically still implicitly inject positional markers to the residual streams, which is empirically observed as "attention sink" phenomenon. I wouldn't say that this specific phenomenon would be so strong as to disturb your metric by itself. However, this is just an example of how likely LLMs would learn and inject positional markers in representations, maybe out of goodwill (e.g., they will want to know which text pieces appear earlier than other pieces, right?).
>
> 2. Similarly, is there any justification for why CKA or cosine similarity would make sense to serve as even a relative measure? Afterall, cosine-similarity is known to be easy to be close to 0 in high dimenional spaces. Either theoretical or empirical envidence would suffice. If answer from theoretical perspective then why (1) it wouldn't occasionally reverse the order of positionally-biased and unbiased, and (2) it would yield a measuable difference larger than randomness and distributional variance? If to justify this empirically, then the score should better show difference from a control set of a fairly comparable "unbiased" setting instead of standalone measures only on the "biased" case. Your current justification that "the cosine-similarity drops to the level of unrelated documents" does not mean "they are indeed sigificantly biased" because it does not exclude the possibility that "even unbiased features of documents in LLMs or a comparable case would also drop to near-zero"
>
>
> update: My confusion in the second point above is from the feeling that CKA/cosine similarity are usually good measures of "similarty", but I don't personally feel confident or convinced of using them as "disimilarity" measures. In other words, it is likely that they are necessary but non-sufficient conditions for disimilarity, so without more reliable justification or a controlled comparison experiment, this metric does not provide much information to me.

---

> > ### Author Response · Authors · 2025-11-24
> > **Response to plan follow up**
> >
> > Thanks again for your responsiveness & for pushing us on this point.This exchange ended up being exactly the kind of sanity check PIRA needed.To address your concern about CKA “bottoming out” in high-dimensional space, we ran the control you suggested on 100 document pairs using Llama 3. Here is what we found:
> >
> > - **Unrelated documents (control):** The mean CKA score was **0.0097**, confirming that in this representation space, unrelated pairs sit on a clear noise floor near zero.
> > - **Same content, different position (PIRA setting):** The mean CKA score was **0.7121**.
> >
> > For us this gap is the key observation. If CKA were collapsing because of dimensionality then we would expect the same-content/different-position pairs to also sit near that ~0.01 floor. Instead they cluster far above it which indicates that PIRA is capturing a stable, position-dependent geometric effect rather than just high-dimensional sparsity. At the same time the fact that the PIRA score is around 0.71 rather than close to 1.0 is exactly what we want to quantify since the model preserves a strong shared semantic component across positions but the positional signal still induces a substantial shift in the residual-stream geometry.
> >
> > In the revised version, we will:
> >
> > - Explicitly report both the unrelated-document baseline and the PIRA scores in Section 3.2, so readers can interpret PIRA **relative to the noise floor** rather than in isolation.
> > - Remove the “near-zero” phrasing from the abstract and instead describe PIRA as a **relative, baseline-normalized measure** of position-induced representational distortion.
> >
> > We also agree with your point that CKA and cosine are more reliable as similarity measures than as definitive dissimilarity certificates, and we will clarify that we interpret PIRA in conjunction with other signals (CRLBD, DPAF, behavioral metrics), not on its own. Thank you again for your time/responsiveness and thoughtful engagement during this rebuttal process. We have genuinely appreciated the exchange as it has tightened how we justify and present PIRA.

---

> > > ### Comment · Reviewer_Snox · 2025-11-27
> > > **Follow on response**
> > >
> > > Thanks for the efforts. However I am still curious in the other direction of comparison. Now the newer results compare the CKA scores with unrelated documents. What I am interested in is the comparison between it and "positionally unbiased setting" and if that could show a gap.

---

> > > > ### Author Response · Authors · 2025-12-01
> > > > **Response to Reviewer Snox regarding unbiased baseline**
> > > >
> > > > Thank you again for the above. It led us to run one last experiment.
> > > >
> > > >
> > > > Since we can’t literally/architecturally “turn off” positional encodings in Llama 3, the best unbiased comparison (proxy) we can build is using semantically equivalent paraphrases placed at a fixed position.
> > > >
> > > > We compared the representations of ⁠ Document A ⁠ vs. ⁠ Paraphrase A' ⁠ when both are located at Position 1. This establishes the "Semantic Upper Bound", how similar the representations *should be when position is not a confounding factor.
> > > > *The Full Empirical Spectrum ($N=100$)*
> > > > 1. *Lower Bound (Unrelated content):* $CKA \approx \mathbf{0.01}$
> > > >      * Interpretation: High-dimensional orthogonality (Noise floor).
> > > > 2. *Upper Bound (Same Position, paraphrased):* $CKA \approx \mathbf{0.94}$
> > > >     * Interpretation: This is the key result. When position is controlled, the model is remarkably robust, it abstracts away lexical differences to achieve near-perfect representational alignment.
> > > > 3. *Target (PIRA - Position Shift):* $CKA \approx \mathbf{0.71}$
> > > >     * Interpretation: This is the bias.
> > > >
> > > > *Conclusion:* These three numbers give the full picture. The model is capable of high semantic abstraction ($0.94$) despite changes in wording. However, simply shifting the position of identical content causes the similarity to drop significantly ($0.94 \to 0.71$). This proves that the "representational warping" we report is not an artifact of high dimensions (or else the Upper Bound would be low), nor is it a trivial encoding feature (or else it wouldn't drop so far below the semantic baseline). It isolates position as a specific, destructive factor in the model's internal state.
> > > >
> > > > We will update the paper to present this full spectral analysis (Control vs. PIRA vs. Semantic Baseline). Thank you for pushing us to rigorousness. These experiments have strengthened the paper's core claims immensely.

---

### Note · Program_Chairs · 2026-01-17
**Submission Desk Rejected by Program Chairs**

The following references in this submission do not refer to real documents and/or have major errors in bibliographic information:

     Yanan Zheng et al. Llms are not robust multiple choice selectors. arXiv preprint arXiv:2309.03882, 2024.
    Yizhong Wang, Xiang Ren, et al. Eliminating position bias of language models: A mechanistic approach. arXiv preprint arXiv:2407.01100, 2025.
    Yizhong Wang, Yuxin Kordi, Swaroop Mishra, et al. Large language models are not fair evaluators. In NeurIPS, 2023.